# The SMHI Large Ensemble (SMHI-LENS) with EC-Earth3.3.1

Klaus Wyser[1], Torben Koenigk[1,2], Uwe Fladrich[1], Ramon Fuentes-Franco[1], Mehdi Pasha Karami[1], Tim Kruschke[1]

[1]Rossby Centre, Swedish Meteorological and Hydrological Institute (SMHI), 601 76 Norrköping, Sweden
[2]Bolin Centre for Climate Research, Stockholm University, 106 91 Stockholm, Sweden

*Correspondence to*: Klaus Wyser (klaus.wyser@smhi.se)

**Abstract.** The Swedish Meteorological and Hydrological Institute used the global climate model EC-Earth3 to perform a large ensemble of simulations (SMHI-LENS). It consists of 50 members, covers the period 1970 to 2100 and comprises the SSP1-1.9, SSP3-3.4, SSP5-3.4-OS and SSP5-8.5 scenarios. Thus, it is currently the only large ensemble that allows for analyzing the effect of delayed mitigation actions versus no mitigation efforts and versus earlier efforts leading to similar radiative forcing at year 2100. We describe the set-up of the SMHI-LENS in detail and provide first examples for its application. The ensemble mean future changes of key variables in atmosphere and ocean are analyzed and compared against the variability across the ensemble members. In agreement with other large ensemble simulations, we find that the future changes in the near surface temperature are more robust than those for precipitation or sea level pressure. As an example for a possible application of the SMHI-LENS, we analyse the probability of exceeding specific global surface warming levels in the different scenarios. None of the scenarios is able to keep global warming in the 21st century below 1.5 ˚C. In SSP1-1.9 there is a probability of approximately 70 % to stay below 2 ˚C warming while all other SSPs exceed this target in every single member of SMHI-LENS during the course of the century. We also investigate the point in time when the SSP5-8.5 and SSP5-3.4 ensembles separate, i.e. when their differences become significant, and likewise when the SSP5-3.4-OS and SSP4-3.4 ensembles become similar. Last, we show that the time of emergence of a separation between different scenarios can vary by several decades when reducing the ensemble size to 10 members.

## 1 Introduction

Global climate is changing due to anthropogenic greenhouse gas emissions (Stocker et al. 2013), and global mean temperature has already increased by more than 1 °C compared to pre-industrial temperature levels. However, the observed temperature time series show large variations on top of the warming trend, resulting in periods of up to decades with reduced or accelerated temperature increase. Changes in climate forcing parameters such as solar irradiance or aerosols and internal climate variability contribute to this observed variability. Especially at the regional level, internal variability leads to large uncertainties on time scales up to several decades (e.g. Hawkins and Sutton 2009, Hawkins 2011). Long-term station observations at different places across Europe show large internal variability of temperature (Moberg et al. 2000). Observations from Uppsala in Sweden, Europe's longest continuous temperature time-series, show that 30-year mean winter

temperature has varied by several degrees between different observed 30-year periods (Moberg and Bergström 1997). Internal climate variability contributes also to uncertainties in future climate projections. At the regional scale, the variability of temperature and particularly precipitation and atmospheric circulation can be as large or even larger as the trends for several decades ahead (Knutti and Sedlacek 2012, Hawkins and Sutton 2012, Deser et al. 2012, Deser et al. 2014, Fischer et al. 2014, von Trentini et al. 2019, Suarez-Gutierrez 2018, Bengtsson and Hodges 2019, Rondeau-Genesse and Braun 2019, Koenigk et al. 2020). In recent studies with large ensembles of CMIP5 models Maher et al. (2020) show that temperature trends in the near-future are largely dominated by internal variability. In agreement with this, Marotzke (2019) found that internal variability masks most of the effects of an efficient implementation of the Paris Agreement until year 2035.

The main source for internal atmospheric variability in middle and high latitudes is the variability of annular modes of circulation (Deser et al. 2012, Horton et al. 2015). Dai and Bloecker (2019) identified the Inter-decadal Pacific Oscillation and Arctic sea ice as main sources for internal climate variability. To robustly distinguish trends in precipitation and atmospheric circulation due to greenhouse gas emissions from internal variability and to better cover the range of possible future climate paths and their extremes in future climate projections, a large ensemble of climate model simulations is necessary. Large ensemble simulations with global coupled models are therefore an important tool. Already in the context of the Coupled Model Intercomparison Project Phase 5 (CMIP5) several global modelling centers performed large ensembles (LENS); 30 members and more have been performed with the following global climate models: CanESM2 (Kirchmeier and Young 2017), MPI-ESM1.1 (Maher et al. 2019), CSIRO-MK3-6 (Jeffrey et al. 2013) and CESM1 (Kay et al. 2015). The MPI Grand Ensemble (MPI-GE) consists of historical and different future projection simulations (rcp26, rcp45, rcp85) while the other ensembles cover parts or the entire historical period and focus on the rcp8.5-scenario. Lehner et al. (2020) used all of these single model LENS to analyse the contribution of internal variability to uncertainties in future climate change. They found that these LENS-simulations provide a good representation of the entire CMIP5 model diversity in many situations.

Also for CMIP6, a few modelling centres have already performed LENS simulations. Ensemble simulations with 30 or more members for at least the historical time period have been performed with CanESM5 (Swart et al. 2019), CNRM-CM6-1 (Voldoire et al. 2019), GISS-E2-1-G (Kelley et al. 2020), IPSL-CM6A-LR (Boucher et al. 2020), NorCPM1 (Bethke et al. 2019), MIROC6 (Tatebe et al. 2019), and SPEAR (Delworth et al. 2020).

An overview on all existing larger ensemble simulations is provided by the Multi Model Large Ensemble Archive (https://www.cesm.ucar.edu/projects/community-projects/MMLEA/ (last accessed 2021-05-09), Deser et al. 2020), which is part of the US CLIVAR Working Group on Large Ensembles. A large number of interesting studies based on large ensembles has already been published (see https://esd.copernicus.org/articles/special_issue1037.html (last accessed 2021-05-09) and Maher et al 2021 for an excellent overview). Examples for the use of large ensembles are studies of the internal variability to obtain robust estimates of extremes (e.g. Kirchmeyer-Young et al. 2017, Haugen et al. 2018) as we intend to do in future studies; or the disentanglement of internal variability and scenario uncertainty (e.g. Marotzke 2019, Lehner et al. 2020, Maher et al. 2020) that is important for the detection of a forced climate change.

In this study, we present and describe the Swedish Meteorological and Hydrological Institute Large Ensemble (SMHI-LENS), performed with the EC-Earth3 model. The simulations follow the CMIP6-protocol (Eyring et al. 2016) and a particular focus is on the effect of mitigation actions for climate change. Thus, the SMHI-LENS comprises the SSP1-1.9, SSP3-3.4, SSP5-3.4-OS and SSP5-8.5. With the exception of SSP5-8.5 these are all Tier-2 scenarios from ScenarioMIP (O'Neill et al. 2016). The wider EC-Earth consortium has already started to contribute ensemble members to the Tier-1 scenarios and it is planned to extend the SMHI-LENS with the Tier-1 scenarios provided access to sufficient computing resources. The Tier-2 scenarios done here are an important extension that allow e.g. for analyzing the effect of delayed mitigation actions versus no mitigation efforts (SSP5-3.4-OS versus SSP5-8.5) and versus earlier efforts leading to similar radiative forcing at year 2100 (SSP5-3.4-OS versus SSP3-3.4). To our knowledge, to date no LENS simulations of SSP4-3.4 and SSP5-3.4-OS exist. The inclusion of SSP1-1.9 allows for analyzing in detail the climate change signal in a world following roughly the Paris agreement. The effect of an overshoot in the forcing has been investigated previously in a smaller ensemble with the CESM model based on CMIP5 forcing (Sanderson et al. 2018). Sanderson et al. (2017) find some impact on land temperature at the time when the difference between the overshoot and steadily increasing forcing are largest, yet the differences are not significant at the gridpoint level. The differences in sea level and Arctic sea-ice are found to be not significant even during the overshoot, and there is no evidence for a long-term climate impact of the overshoot at the end of the 21st century and beyond.

Another motivation of performing the SMHI-LENS was to provide boundary conditions for downscaling simulations with regional models that include a large part of the internal variability. Several studies with regional climate models with boundary forcing from a single model large ensemble have already been published (e.g. Leduc et al. 2019, von Trentini et al. 2019, Mankin et al. 2020, Böhnisch et al. 2020, von Trentini et al. 2020). Through clever selection of members from the large ensemble almost the entire range of uncertainty due to internal variability within a certain region or for certain processes can be spun up with a relatively small number of regional downscalings.

## 2 Model and simulations

### 2.1 Model description

The SMHI-LENS was generated with EC-Earth3 version 3.3.1 that has been used in the GCM configuration that comprises IFS cy36r4 for the atmosphere and NEMO3.6 including the sea ice model LIM3 for the ocean (Döscher et al. 2021). The atmosphere model uses the spectral truncation T255 combined with a linearly reduced Gauss grid with a resolution of about 80 km (N128), and the ocean model uses the tri-polar ORCA1 grid with a 1-deg resolution over large parts of the globe and a mesh refinement at the equator. In the vertical there are 91 levels in the atmosphere with the top level at 1 Pa, and 75 layers in the ocean with an upper level of about 1 m and 24 levels distributed over the uppermost 100 m.

The time step is 45 minutes in the atmosphere and ocean, and the coupling between atmosphere and ocean is done at every time step.

The forcing for all simulations is identical to what has been used for making the EC-Earth3 contribution to CMIP6. In particular we use the GHG concentrations, solar radiation, stratospheric ozone concentration and stratospheric aerosols (volcanoes) for CMIP6. The anthropogenic aerosol forcing MACv2-SP (Stevens et al. 2017) has been implemented in EC-Earth3 and is used in combination with a climatological pre-industrial aerosol background. Time varying land use is accounted for by using pre-computed vegetation cover, type and leaf area indices; these forcings have been obtained from previous historical and scenario simulations with EC-Earth3-Veg, the model configuration that includes the dynamic vegetation model LPJ-Guess.

## 2.2 Initial conditions

To create the set of initial conditions for SMHI-LENS we start from 6 members (r1-3, r7-9) of the historical experiment for CMIP6 that was done with EC-Earth3-Veg.  From each of the 6 members we branch off breeding simulations on Jan 1, 1970. These 6 breeding simulations are each run for 20 years with constant forcing. From these 6 breeding runs we then select 50 initial states for the atmosphere and the ocean as initial conditions for the large ensemble (Table 1). The initial date for each member of the large ensemble is set to Jan 1. 1970.

To check the ensemble spread after initialization and whether it captures the full intra-model variability, we compare the ensemble spread in SMHI-LENS in year 1970 (annual mean) to EC-Earth3 historical simulations for CMIP6 (23 members) which have been integrated independently for already 120 model years at this point (Fig. 1). There are differences between the two ensembles but these are not significant (at the 5% level) neither for the means nor for the variances of the two ensembles and therefore the two ensembles can be considered independent samples of the same distribution.

## 2.3 Simulations

The 50 members of the historical ensemble were started in 1970 from the 50 initial conditions, using the forcing provided for the historical experiment for CMIP6. The historical simulations run until the end of 2014 followed by several scenarios that cover the 2015-2100 period with forcings according to ScenarioMIP. Each of the scenarios comprises 50 members that start from the end of the corresponding member of the historical experiment. The following scenarios are included in the large ensemble:

- SSP5-8.5 is a high-end scenario that yields a strong warming signal, marking the upper end of a plausible evolution of the climate
- SSP5-3.4-OS is an overshoot scenario with a strong warming until 2040 (using the same forcing as SSP5-8.5 until 2040), followed by a curbing and net-negative emissions after 2060 resulting in a radiative forcing of 3.4 W m-2 in 2100. The difference between SSP5-8.5 and SSP5-3.4-OS will tell about the efficacy of mitigation measures that set in around the mid-century. Following the CMIP6 protocol, the SSP5-3.4-OS experiment branches off from the SSP5-8.5 experiment in 2040 which means results for SSP5-3.4-OS are only available for the 2040-2100 period.

- SSP4-3.4 also has a radiative forcing of 3.4 W m-2 in 2100, but without the peak and decline of SSP5-3.4-OS. Differences between these two scenarios can tell us about the impact of a previous overshoot and possible non-reversible effects when the forcing at the end of the century is similar.

- SSP1-1.9 is the low-end scenario addressing the needs of the Paris Agreement to reach the 1.5 degree warming level, marking the lower end of a plausible evolution of the climate.

All these scenarios except SSP5-8.5 are from Tier-2 of ScenarioMIP. The wider EC-Earth community is planning to provide between 20 and 30 members of the Tier-1 scenarios, and therefore it was considered more valuable to extend the EC-Earth contribution to CMIP6 with Tier-2 scenarios. Furthermore, the selection of scenarios for the large ensemble was guided by

135 questions about the impact of mitigation and overshoot. Nevertheless, the low- and high-end scenarios of the SMHI-LENS span the full range of possible futures.

## 2.4 Data output

Limitations on storage capacity do not allow us to save the full model output as it has been done for CMIP6. Instead we select a subset of variables from ocean and atmosphere, and save only daily and monthly means. Tables 2 and 3 list the

140 variables for atmosphere and ocean, respectively. All data from the large ensemble are CMIP6 compliant and are available from any ESGF data portal as part of the CMIP6 data holding. Realisation_id's r101 to r150 from the EC-Earth3 model have been reserved for the large ensemble.

The limited output does not allow for any in-depth analysis of extreme events such as strong storms or an extreme precipitation event on sub-daily timescales. We therefore plan to re-run selected periods with full output and, for this

purpose, have saved the full model state on Jan 1 of each year, for each member and for each scenario.

## 3 Results

The aim of this work is to provide an overview of SMHI-LENS and we therefore focus only on main characteristics of major variables. To benefit from the large number of ensemble members, we not only look at ensemble means but also at the ensemble spread as a measure of the internal variability, both in global mean timeseries as well as in the analysis of regional

climate change patterns. More detailed studies with the data from SMHI-LENS are in preparation.

### 3.1 Timeseries

Timeseries of global annual mean temperature and precipitation are displayed in Fig. 2, together with timeseries of AMOC and the Arctic minimum sea ice extent. The ensemble spread is illustrated by the shaded area that shows the full spread, minimum to maximum of the ensemble. The scenarios continue the historical experiment after 2014 with little differences

among the different scenarios. They start diverging first around year 2040 for three out of four variables considered here. The exception is the Arctic sea ice minimum where the reduction in SSP5-8.5 is stronger than in the other scenarios already

after year 2030 (Fig. 2d). The temperature timeseries (Fig 2a) shows the anticipated warming of the different scenarios with a strong warming signal in SSP5-8.5 that keeps increasing throughout the 21st century while SSP1-1.9 first overshoots slightly and then stabilises around the mid-century at a level only slightly higher than the present-day climate. Increasing temperatures lead to a more vigorous hydrological cycle with increased global precipitation (Fig 2b) and a decrease in the Arctic sea ice minimum (Fig 2d). We also find a distinct impact on the AMOC that first weakens compared to present-day conditions but partly recovers in all scenarios except for the high end SSP5-8.5 scenario (Fig 2c). The ensemble spread in AMOC is high for the historical period but shows a reduction in all scenarios after the middle of the 21st century. The AMOC is closely connected to the oceanic convection in the Labrador Sea and its variability (Brodeau and Koenigk 2016, Koenigk et al. 2021). The convection in the Labrador Sea becomes weaker in all scenarios until the middle of the 21st century (not shown), which reduces the ensemble mean and ensemble spread of AMOC. The mitigation measures as represented in SSP5-3.4-OS lead to more or less immediate impacts on global mean temperature and precipitation while an imprint onto the AMOC becomes visible with a delay of approx. 20 years.

## 3.2 Regional patterns

The ensemble mean annual mean 2m air temperature (tas) averaged over 1995-2014 shows the well-known north south gradients with minimum values below -20°C in the polar regions and up to 30°C in the tropics (Fig. 3 a). The typical discrepancies from the zonality, for example the tongue of warm air in the northeastern North Atlantic and North Pacific and colder tas over the parts of the northern hemispheric continents are well reproduced. Details on biases in the mean climate in EC-Earth3 are provided by Döscher et al. (2021). The standard deviation of tas, averaged over 1995-2014, across model members shows substantial internal variability with largest variability near the ice edges of the North Atlantic Arctic sector where one standard deviation reaches values of 3 K and higher. Also, mid and high latitude regions of the northern hemispheric continents and the ice regions around Antarctica experience high internal tas variability. In subtropical and tropical areas, one standard deviation of tas variability is generally below 0.5 K.

The ensemble mean temperature change until the middle of the 21st century shows a clear Arctic amplification with the largest warming rates in regions where even winter sea ice disappears, especially in the Barents and Kara Seas. Here, warming exceeds 5 K in all scenarios until 2040-2059, and reaches even more than 10 K in the SSP5-3.4-OS and SSP5-8.5 scenarios. Over the continents, the warming is generally larger than over the oceans, and is smallest over the mid-latitude oceans of the southern hemisphere with warming rates below 1 K.

The general warming patterns are similar in the different scenarios. The warming until 2040-2059 is somewhat more pronounced in SSP5-3.4-OS and SSP5-8.5 compared to SSP4-3.4 and SSP1-1.9. The difference between the scenarios increases until the end of the century. While especially SSP5-8.5 shows an accelerated tas increase until 2080-2099, tas in SSP1-1.9 does not increase any more compared to 2040-2059. The tas increase in SSP5-3.4-OS is small after 2040-2059 compared to SSP5-8.5, and is similar to the one in SSP4-3.4 by the end of the century. This shows the impact of the strongly decreasing greenhouse gas emissions in SSP5-3.4-OS after 2040.

Figs. 3 k-n display the ratio between mean tas change and internal variability. The ensemble mean tas change is divided by one standard deviation of tas change across the ensemble members. The tas change has been calculated separately for each ensemble member by subtracting tas in 2040-2059 (2080-2099) from 1995-2014 in the same ensemble member. If this signal-to-noise ratio is 2 – meaning that the change signal exceeds two standard deviations of variability of tas change - would indicate that around 97.5 % of all members show a warming signal. The spatial pattern of one standard deviation of

variability of tas change across members (not shown) is very similar to the standard deviation of tas in 1995-2014 (Fig. 3 b) but the amplitude is slightly higher, particularly for the change until 2040-2059 (not shown). The variability of tas changes until 2080-2099 is slightly smaller than until 2040-2059 (not shown) because of compensating decadal scale periods of internal variability (Koenigk et al. 2020).

The ratio between mean tas change until 2040-2059 in SSP1-1.9 and variability exceeds 2 for most regions of the world

except for the northern North Atlantic and the Southern Ocean around Antarctica. Also, parts of western and northern Europe show a comparatively small ratio. At the end of the century, the ratio between mean tas change and variability of change in SSP1-1.9 increases in many regions, mainly due to the larger signal under stronger forcing. Under the SSP5-8.5 scenario, the ratio is substantially higher than in SSP1-1.9 - only the subpolar gyre regions and Nordic Seas show a ratio between mean change until 2040-2059 and variability of change below 2. At the end of the century, the ratio is 2.5 - 5 in the

northern North Atlantic and exceeds 10 in most areas of the world, showing the clear dominance of the change signal over the variability.  The small signal to noise ratio in the northern North Atlantic can be linked to the reduction in the AMOC (compare Fig. 2c) and the related northward heat transport into the North Atlantic.

The spatial annual mean precipitation (pr) distribution in EC-Earth3 is dominated by low values in polar regions and subtropical regions and high pr in the tropics, in maritime mid-latitude regions and along mountain ranges (Fig. 4 a). EC-

210 Earth3 generally well reproduces the observed pr pattern but it shows a double intertropical convergence zone bias, dry biases over some parts of central and western Eurasia and wet biases over parts of the polar regions and the subtropical oceans of the southern hemisphere (for details see Döscher et al. 2021).

The largest variability of annual mean pr averaged over 1995-2014 across model members occurs in the tropics, along the Gulf Stream and North Atlantic Current as well as in the subpolar gyre and along the ice edges of the North Atlantic Arctic

sector. These are also some of the areas that show the largest projected pr changes: pr is significantly increased over the tropical oceans, except for the tropical Atlantic where both regions with increased and decreased pr occur, and in the polar regions, particularly along the ice edges. Over the tropical land regions, the signal is noisy with both positive and negative signals. The Sahel zone shows increased pr. Further, it generally gets wetter over most of mid and high latitudes. In most of the subtropical ocean regions of the southern hemisphere, pr is significantly decreased. The change pattern agrees well

across the different emission scenarios. As for tas change, the amplitude of pr change until 2040-2059 is somewhat larger in SSP5-3.4-OS and SSP5-8.5 compared to SSP4-3.4 and SSP1-1.9. Until the end of the century, P-changes substantially increase in SSP5-8.5 and the differences across scenarios become more pronounced. SSP4-3.4 shows also further amplified

pr changes while the additional changes in SSP5-3.4-OS and particularly in SSP1-1.9 are small. As for tas, SSP5-3.4-OS shows similar P-changes as SSP4-3.4 at the end of the century.

Note that we discuss absolute values of P-change and not values in percentage. In percentage, the largest changes occur over the northern hemispheric polar regions with up to 50-100% increase in SSP5-8.5 at the end of the century compared to 1995-2014 (not shown). Here, the ratio of the mean P-change versus variability of the trend across members (Figs. 4k-n) is largest and exceeds 2 in SSP1-1.9 in 2040-2059 and reaches up to 10 in SSP5-8.5 in 2080-2099. In SSP5-8.5, the mean P-change dominates over the variability in southern polar and tropical regions as well. However, in many mid and sub-tropical regions,

the variability is larger than the mean change signal in all scenarios and even for changes until the end of the 21st century.

The atmospheric circulation and its potential future changes are highly relevant for the spatial distribution of tas and pr and their future changes. To characterize the circulation, we analyse the sea level pressure (psl, Fig. 5). The mean psl in the period 1995-2014 represents well the observed psl and biases in EC-Earth3 are between -1 and +1 hPa in most areas of the world (Döscher et al. 2021). In the North Pacific, the Aleutian Low is slightly too pronounced, in the subpolar North

Atlantic, psl biases of up to 2 hPa exist and over parts of the Antarctic, psl is up to 2 hPa too high compared to ERA5-reanalysis data. The standard deviation in these regions is considerably smaller than the differences to ERA5 and therefore the biases are consistent among the majority of ensemble members in the sense that they have the same sign. The psl variability across members is generally largest in mid and high latitudes of both hemispheres, and one standard deviation of psl variability reaches here up to around 1 hPa (Fig. 5 b). In the tropics, the psl variability is small and one standard deviation

is below 0.2 hPa.

The change of psl until 2040-2059 is small and not significant at the 95% significance level in many areas. The change in psl is asymmetric, more pronounced in southern hemispheric mid-latitudes and some subtropical and tropical regions where changes are positive and can reach up to 1 hPa in SSP1-1.9 and SSP3-3.4-OS and up to 1.5 hPa in SSP5-3.4-OS and SSP5-8.5. Over the tropical Pacific Ocean, the mean zonal circulation weakens in the SMHI-LENS future scenarios with slightly

increased psl over the west Pacific, in comparison to the psl over the east side of the Pacific, therefore showing an eastward shift of the Walker Circulation, and negative values of the Southern Oscillation Index (SOI: difference in psl between Tahiti in the eastern Pacific and Darwin in northern Australia). These future changes in the mean circulation observed in all the scenarios, show a very similar pattern to the positive phase of El Niño Southern Oscillation variability, when the Walker circulation weakens and the rising branch over the Maritime Continent shifts to the east in comparison to neutral conditions.

A shift towards more El Niño-like conditions under global warming agrees with the majority of previous CMIP3 and CMIP5 projections (Vecchi et al. 2006, 2007; Bayr et al. 2014) although Kohyama et al. (2017) find that also a more La Niña–like trend could be a physically consistent response to warming.

In polar regions, psl generally decreases in all scenarios in both hemispheres. The spatial psl change pattern remains similar in 2080-2099 compared to 2040-2059. However, as for tas and P, the amplitude of psl change in SSP5-8.5 is strongly

enhanced compared to the period 2040-2059. In polar regions, psl is reduced by more than 3 hPa and it is increased by up to 3 hPa in southern hemisphere mid-latitudes. In contrast to the other SSPs, SSP5-8.5 shows significantly increased psl in most

northern hemispheric ocean regions as well. Despite these larger changes until 2080-2099 in SSP5-8.5, the variability strongly dominates over the mean change in all polar regions and in most of Eurasia, North Africa and North America as well as over the North Atlantic. In SSP1-1.9, the mean change is only robust across model members in larger parts of the area between 10° N and 40° S.

### 3.3 Probability of exceeding specific surface warming levels

An important question of climate adaptation is the likelihood for passing a specific surface warming level (SWL). The large ensemble allows for a quantitative estimate of the probability of surpassing a given temperature. It is common practice to express the warming relative to pre-industrial levels, in other words the difference between the global mean temperature in the future scenarios and the global mean pre-industrial temperature. The pre-industrial temperature is computed as the ensemble mean of 23 realisations of the historical EC-Earth3 experiments for the 1850-1870 period that have been published on the ESGF. For each year we then compute the fraction of the SMHI-LENS members that exceed a given warming threshold.

The probability for exceeding three different SWLs in the four scenarios is shown in Fig 6. All members of SSP5-8.5 exceed SWL3 after 2060 (Fig 6a). SSP5-3.4-OS that branches off from SSP5-8.5 after 2040 reaches only about 20 % probability for exceeding SWL3 during 2060-2080 and shows lower probability thereafter, demonstrating clearly the impact of the mitigation that is underlying this specific scenario. SWL2 and SWL1.5 are tightly linked to the Paris Agreement that aims at avoiding warming above 2 or 1.5 degrees. Our results with the 4 scenarios used here reveal that only SSP1-1.9 is likely to keep the warming below 2 degrees (Fig 6b). There still is an almost 40% probability for exceeding SWL2 even in SSP1-1.9 around the middle of the century after which the probability becomes lower again. In the other scenarios the likelihood to pass SWL2 reaches 100% around year 2040 in SSP5-8.5 and about 20 years later in SSP4-3.4. The more ambitious 1.5 degrees warming target cannot be reached by any of the scenarios used here, the likelihood to exceed SWL1.5 reaches 100% before 2040 with little variation among the scenarios which makes them almost indistinguishable in Fig 6c. The future analysis of SMHI-LENS will include a more thorough investigation of the impact from an overshoot in the climate trajectory.

### 3.4 Separation of scenarios

Experiments SSP5-3.4-OS and SSP4-3.4 both end with an approximately equal climate forcing of 3.4 W m-2 in 2100 yet their pathway is rather different (Fig. 2) with SSP5-3.4-OS showing an overshoot in the middle of the century while SSP4-3.4 shows a constantly increasing temperature response. The question arises if and when SSP5-3.4-OS becomes different from SSP5-8.5, and when SSP5-3.4-OS approaches and becomes similar to SSP4-3.4. To answer these questions, we compare the ensembles of the annual mean temperature from each of these experiments and decide when and where the differences between the ensembles are statistically significant with help of a Student's t-test. The t-score between two ensembles is calculated as

$$t = \frac{|m_1 - m_2|}{\sqrt{\frac{s_1^2}{n_1} + \frac{s_2^2}{n_2}}} \qquad\qquad (1)$$

where $m$ denotes the ensemble mean, $s$ the std deviation and $n$ the number of members in each ensemble. The difference between the two ensembles with 50 members each is significant at the 95% level when t exceeds $t^*(0.95,49) = 2.009$ for the two-sided 95% confidence level and 49 degrees of freedom.

We apply Eq (1) to the annual temperature means of the SSP5-8.5 and SSP5-3.4-OS experiments to compute the t-score in each gridpoint and for each year. The t-scores are then smoothed with a 5-yr running mean. Fig. 7a displays the year after

295 which the smoothed t-scores become larger than $t^*(0.95,49)$, indicating the year after which the differences between SSP5-8.5 and SSP5-3.4-OS have diverged enough for their difference being statistically significant. Similarly, Fig. 7b shows the year after which the difference between SSP5-3.4-OS and SSP4-3.4 is not significant any longer, telling when the two scenarios have converged.

The differences in annual mean temperature between SSP5-8.5 and SSP5-3.4-OS emerge in most regions between 2050 and

300 2060, with the exception of Antarctica and the Southern Ocean, Africa south of the Sahara, India and central Australia where the differences become significant after 2060 (Fig 7a). The temperature differences between SSP5-3.4-OS and SSP4-3.4 show larger spatial variability (Fig 7b). There is a hint of a North-South gradient in the year when the difference between these two scenarios ceases to be significantly different. In the Northern Hemisphere the last year with a significant difference occurs during the 2060-2080 period in most gridpoints, with notable exceptions in Northern Canada and Greenland. In the

305 Southern Hemisphere the temperature differences are significant until 2080-2100 over large areas of the Oceans, Africa and Antarctica. Over South America and Australia the temperature difference between SSP5-3.4-OS and SSP4-3.4 ceases to be significant in the 2070-2080 period.

How does this result depend on the ensemble size? The t-score that is used to assess if the temperature differences between 2 scenarios are significant is proportional to the square root of the ensemble size. Furthermore, the $t^*$ value for testing

significance depends on the degrees of freedom that in turn depend on the number of ensemble members. Let us now assume that for each of the scenarios used here we have a hypothetical ensemble with the same mean and variance as the large ensemble, but only 30 (or 10) members. The t-scores for the difference between two scenarios are first scaled by $\sqrt{3}$ ($\sqrt{5}$) and then compared to $t^*(0.95,29) = 2.045$ ($t^*(0.95,9) = 2.262$) to assess significance at the 95% level. This approach reflects the larger uncertainty that follows from the smaller sample size. The results for the time of detection of significant

differences between SSP5-8.5 and SSP5-3.4-OS and the time of cessation of significant differences between SSP5-3.4-OS and SSP4-34 are shown in Figs. 7c and d for 30 members, and Figs. 7e and f for 10 members. Comparing Figs. 7a, c and e we find that the difference between SSP5-8.5 and SSP5-3.4-OS would become detectable about a decade later if the ensemble consisted of only 10 members instead of 30 or 50. The impact of a reduction of the ensemble size is more drastic when it comes to the differences between SSP5-3.4-OS and SSP4-34 (Figs. 7b, d and f). The time of emergence and

cessation of significant differences doesn't differ much between 50 and 30 members, the large differences appear first when

the ensemble size is reduced to 10 members. Many regions and most notably the Northern Hemisphere continental areas do not show any significant temperature differences between these two scenarios during the 21st century if only 10 ensemble members were available. And even in regions where differences between SSP5-3.4-OS and SSP4-3.4 would still be significant, the differences would stop being significant several decades ahead of the time when it happens with 50 or 30 members, thus reducing the period where the two scenarios can be considered to be distinct from each other. This would be a clear drawback for any studies of the impact from the overshoot in SSP5-3.4-OS as the number of available years for such an analysis would be limited. Fig. 7 is a clear example for the need of sufficiently large ensembles when assessing differences between certain scenarios to assess the impacts of mitigation measures.

The analysis of the emergence/cessation of significant differences between different experiments could be expanded to all scenarios, this would however be beyond the scope of the present paper to provide an overview over SMHI-LENS and will be saved for future studies.

## 4 Discussion and conclusions

Here we have presented an overview of the SMHI Large Ensemble that consists of 50 members done with the EC-Earth3 model. We described the process of creating a large set of initial conditions for 1970 starting from 6 members of the ensemble of the historical experiment that in turn had branched off at various points in time from the piControl experiment.

The future projections, following the ScenarioMIP-protocol, have shown the anticipated results: a strong warming with SSP5-8.5, an overshoot in the warming with SSP5-3.4-OS in the middle of the century followed by a negative warming trend towards the end of the century, a continuously increasing warming with SSP4-3.4 reaching the same level of warming as SSP5-3.4-OS towards 2100, and a limited warming with SSP1-1.9. Not surprisingly, the projections in the large ensemble are in line with other CMIP6 results, the advantage of the large ensemble being that it allows us to better quantify the impact of internal variability on the changes and thus derive results subject to reduced uncertainty.

When comparing the mean future change against the variability of the change across the ensemble we have found that the future changes in the near surface temperature are significant almost everywhere but not for precipitation or sea level pressure. This result agrees qualitatively with earlier studies involving large ensembles yet there are regional differences between SMHI-LENS and large ensembles from other models. Deser et al. (2012) show in agreement to our results that the mean temperature change signal is much more robust than pr and psl change signals. For pr and psl, they found similar regions with large and small ratios between mean change and internal variability as this study. Both regions and amplitudes of standard deviation of tas, psl and pr trends agree relatively well with our results. Compared to results from the MPI-ESM1.1 Grand Ensemble (MPI-GE, Maher et al. 2019), the variability of psl changes until the end of the 21st century is comparable in pattern and amplitude as well. However, the mean psl change signal differs somewhat. While most regions show small psl change in MPI- GE similar to SMHI-LENS, Maher et al. (2019) found two areas with stronger responses as in SMHI-LENS: psl increases from Greenland across the northeastern North Atlantic to central and southern Europe and a

strong negative signal over the Bering Sea region. On the other hand, the psl decrease over the Arctic seems to be smaller in MPI- GE.

The slightly reduced internal variability for tas and pr changes until the end of the century (2080-2099) compared to the middle of the century (2040-2059) in SMHI-LENS is in line with findings for Europe by Koenigk et al. (2020) based on the MPI- GE and the CanESM2-Large Ensemble. They linked this reduced internal variability to compensating decadal scale periods of internal variability, which enhance and slow down the mean trend due to greenhouse gas emissions.

An important application for large ensembles is risk assessment; as an example, we analyze the probability for exceeding a
specific warming level in a given scenario. Many impact studies have looked at the effects when a certain warming is passed (e.g. Donnelly et al. (2017), Teichmann et al. (2018), Koutroulis et al. (2018)), but only few studies so far have analysed the probability itself for passing a specific warming level. We show that none of the scenarios used here is able to keep global warming in the 21st century below 1.5 degrees. In SSP1-1.9 there is an approximately 70% probability for the warming to stay below 2 degrees warming while all other SSPs exceed this target during the course of the century. SSP5-8.5 is the only
one of the used scenarios to definitely pass even a 3-degree warming. SSP5-3.4-OS has a 20-40% chance to exceed SWL3 temporarily during the 2050-2090 period, but at the end of the century the risk of warming beyond this threshold is very small. For comparison, based on the CMIP5 model ensemble, Jiang et al. (2016) show that the probability to exceed the 2 °C global warming level before the year 2100 is 26, 86, and 100% for the Representative Concentration Pathways 2.6 (RCP2.6), 4.5 (RCP4.5), and 8.5 (RCP8.5) scenarios, respectively, with the median years of 2054 for RCP4.5 and 2042 for RCP8.5.

To demonstrate the importance of a sufficiently large ensemble we look at the point in time when the differences between the SSP5-8.5 and SSP5-3.4-OS ensembles become significant, and when the SSP5-3.4-OS and SSP4-3.4 ensembles become similar. When assuming that the ensemble would retain the mean and variance but with only 10 members, we show that the time of emergence of a separation between SSP5-8.5 and SSP5-3.4-OS can vary by several decades. The impact of the ensemble size is even more apparent when looking at the time when SSP5-3.4-OS and SSP4-3.4 stop being significantly
different. With 50 members this happens in the 2nd half of the 21st century implying that the overshoot and the gradually increasing scenario really lead to distinct responses in the temperature. With only 10 members the overshoot becomes much less detectable and there are large regions where the temperatures in SSP5-3.4-OS and SSP4-3.4 are indistinguishable.

A major reason for making large ensemble simulations is to assess the natural variability and thereby obtain an estimate for the uncertainty in future projections. It is common practice to take the ensemble spread as an estimate for the natural
variability. However, the internal variability differs between models and thus the ensemble spread will also be different for ensembles created with different models (Lehner et al 2020). The EC-Earth3 model is among the models with highest variability in the piControl run (Parsons et al. 2020) for reasons not yet fully understood. Thus, SMHI-LENS likely has a large ensemble spread which implies that the uncertainty estimates such as confidence intervals get wider. Furthermore, the difference between ensembles for the scenarios needs to be bigger for their clear separation which has an impact on the time
when scenarios are similar or significantly different. The results presented here are thus model dependent and could look different for other large ensembles done with a different model.

The results presented here are just examples for what kind of analyses and risk assessment are possible with a large ensemble. In the future it is planned to extend this kind of work to regional warming signals, the frequency of occurrence of extreme events (e.g. heat waves), detection and attribution studies, and other variables (e.g. precipitation, sea-ice).

**Code availability**

The EC-Earth model is restricted to institutes that have signed a memorandum of understanding or letter of intent with the EC-Earth consortium and a software license agreement with the ECMWF. Confidential access to the code and to the data used to produce the simulations described in this paper can be granted for editors and reviewers; please use the contact form at http://www.ec-earth.org/about/contact.

**Data availability**

All results from SMHI-LENS are available from any ESGF datanode as part of CMIP6, search for <model_id> EC-Earth3 and <variant_label> in the range r101i1p1f1 and r150i1p1f1.

**Author contribution**

All authors contributed to the design of the experiments. KW performed the simulations. TK and KW performed the
400 analyses, prepared the figures and an initial draft of the manuscript. All authors contributed to the discussion of the results and the final manuscript.

**Competing interests**

The authors declare that they have no conflict of interest.

**Acknowledgements**

The computations were enabled by resources provided by the Swedish National Infrastructure for Computing (SNIC) at the National Supercomputer Centre at Linköping University, partially funded by the Swedish Research Council through grant agreement no. 2018-05973. We'd like to thank the editor and two anonymous reviewers for their valuable comments and suggestions that helped us to improve the manuscript.

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

**Tables**

| parent_id of CMIP6 historical EC-Earth3-Veg from which breed experiment branches off at 1970-1-1 | branch_time in breed experiment | SMHI-LENS variant_label_id |
|---|---|---|
| | 1990-1-1 | r\<n+100>i1p1f1 |
| | 1988-1-1 | r\<n+106>i1p1f1 |
| | 1986-1-1 | r\<n+112>i1p1f1 |
| | 1984-1-1 | r\<n+118>i1p1f1 |
| r\<n>i1p1f1 | 1982-1-1 | r\<n+124>i1p1f1 |
| | 1980-1-1 | r\<n+130>i1p1f1 |
| | 1978-1-1 | r\<n+136>i1p1f1 |
| | 1976-1-1 | r\<n+142>i1p1f1 |
| | 1974-1-1 | r\<n+148>i1p1f1 |

Table 1: Relationship between the members of the CMIP6 historical experiment done with EC-Earth3-Veg and the members of the SMHI-LENS. \<n> is the realisation_id (member) of the CMIP6 historical experiment (1-6). The branch_time 1974-1-1 is only used for r1 and r2 of the CMIP6 historical experiment, yielding members 149 and 150 of the large ensemble.

| Short name | Name | Type | Frequency |
|---|---|---|---|
| ta | Temperature | PL | monthly + daily |
| ua | Zonal wind | PL | monthly + daily |
| va | Meridional wind | PL | monthly + daily |
| hus | Specific humidity | PL | monthly |
| zg | Geopotential | PL | monthly + daily |
| tas | 2m temperature | SFC | monthly + daily |

| tasmax | 2m minimum temp | SFC | monthly + daily |
|--------|-----------------|-----|-----------------|
| tasmin | 2m maximum temp | SFC | monthly + daily |
| hurs | 2m relative humidity | SFC | monthly + daily |
| huss | 2m specific humidity | SFC | monthly + daily |
| pr | Total precipitation | SFC | monthly + daily |
| prc | Convective precipitation | SFC | daily |
| prsn | Snowfall | SFC | monthly + daily |
| evspsbl | Evaporation | SFC | monthly |
| sfcWind | 10m wind speed | SFC | monthly + daily |
| uas | 10m wind component | SFC | daily |
| vas | 10m wind component | SFC | daily |
| clt | Total cloud cover | SFC | monthly |
| clwvi | Liquid water path | SFC | monthly |
| clivi | Ice water path | SFC | monthly |
| prw | Precipitable water | SFC | monthly |
| psl | Mean sea level pressure | SFC | monthly + daily |
| snw | Snow water equivalent | SFC | monthly |
| mrro | Runoff | SFC | monthly |
| rsds | SW flux downward | SFC | monthly + daily |
| rsus | SW flux upward | SFC | monthly |

| | | | |
|---|---|---|---|
| rlds | LW flux downward | SFC | monthly + daily |
| rlus | LW flux upward | SFC | monthly |
| hfls | Latent heat flux | SFC | monthly |
| hfss | Sensible heat flux | SFC | monthly |
| rsdt | SW flux downward | TOA | monthly |
| rsut | SW flux upward | TOA | monthly |
| rlut | LW flux upward | TOA | monthly |
| tsl | Soil temperature | Soil | monthly |
| mrso | Soil moisture | Soil | monthly |

Table 2: Saved atmosphere variables. The column labelled "*Type*" indicates if a variable is saved at the surface (SFC), at top of the atmosphere (TOA), in the soil or on pressure levels (PL). Monthly means are saved on 19 pressure levels (plev19 in the CMIP6 tables) and daily means/maxima on 3 pressure levels (plev3).

| *Short name* | *Name* | *Type* | *Frequency* |
|---|---|---|---|
| tos | Sea surface temperature | SFC | monthly |
| zos | Sea surface height | SFC | monthly |
| sos | Sea surface salinity | SFC | monthly |
| siconc | Sea ice cover | SFC | monthly |
| sivol | Sea ice volume | SFC | monthly |
| siu | Sea ice zonal velocity | SFC | monthly |
| siv | Sea ice meridional velocity | SFC | monthly |

| | | | |
|---|---|---|---|
| mlotst | Mixed layer depth | SFC | monthly |
| hfx | Zonal heat flux (vertically integrated) | SFC | monthly |
| hfy | Meridional heat flux (vertically integrated) | SFC | monthly |
| thetao | Temperature | 3-D | monthly |
| so | Salinity | 3-D | monthly |
| uo | Zonal velocity | 3-D | monthly |
| vo | Meridional velocity | 3-D | monthly |

Table 3: As Table 1 but for saved ocean variables. All ocean variables are saved as monthly means. 3-D variables are provided on the native 75 levels of the ORCA1L75 grid.

**Figures**

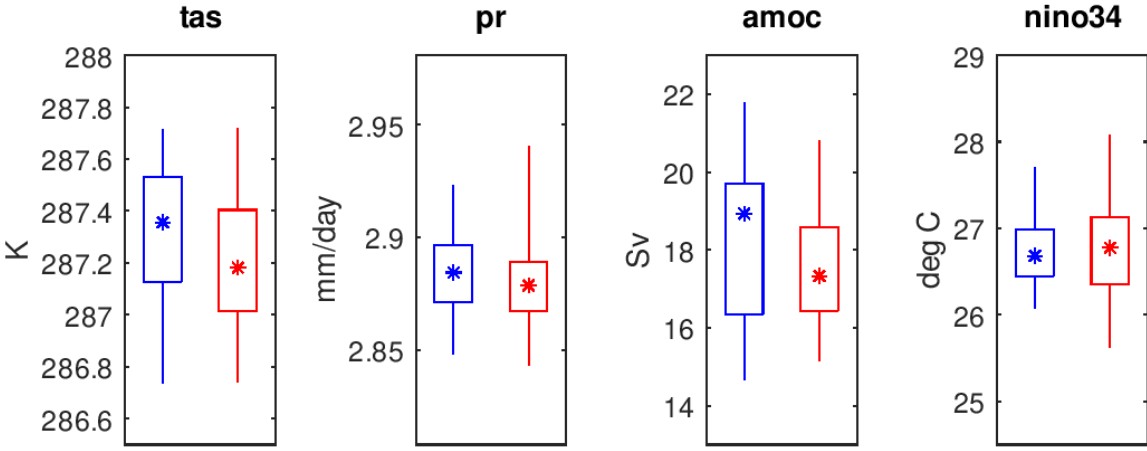

Figure 1: Ensemble spread in the first year after the initialization of the large ensemble (red), compared to the ensemble spread of the regular EC-Earth3 ensemble for year 1970 of the historical experiment for CMIP6 (blue). The whiskers denote the full ensemble spread (min-max), the boxes the 25- to 75-percentile range, and the stars the median of the distribution. Tas and pr are the annual global mean near surface temperature and precipitation, respectively. AMOC is the annual average of the monthly maximum Atlantic Meridional Overturning Circulation at 26.5 deg N. Nino34 is the annual average SST in the Nino3.4 region (5N-5S, 170W-120W).

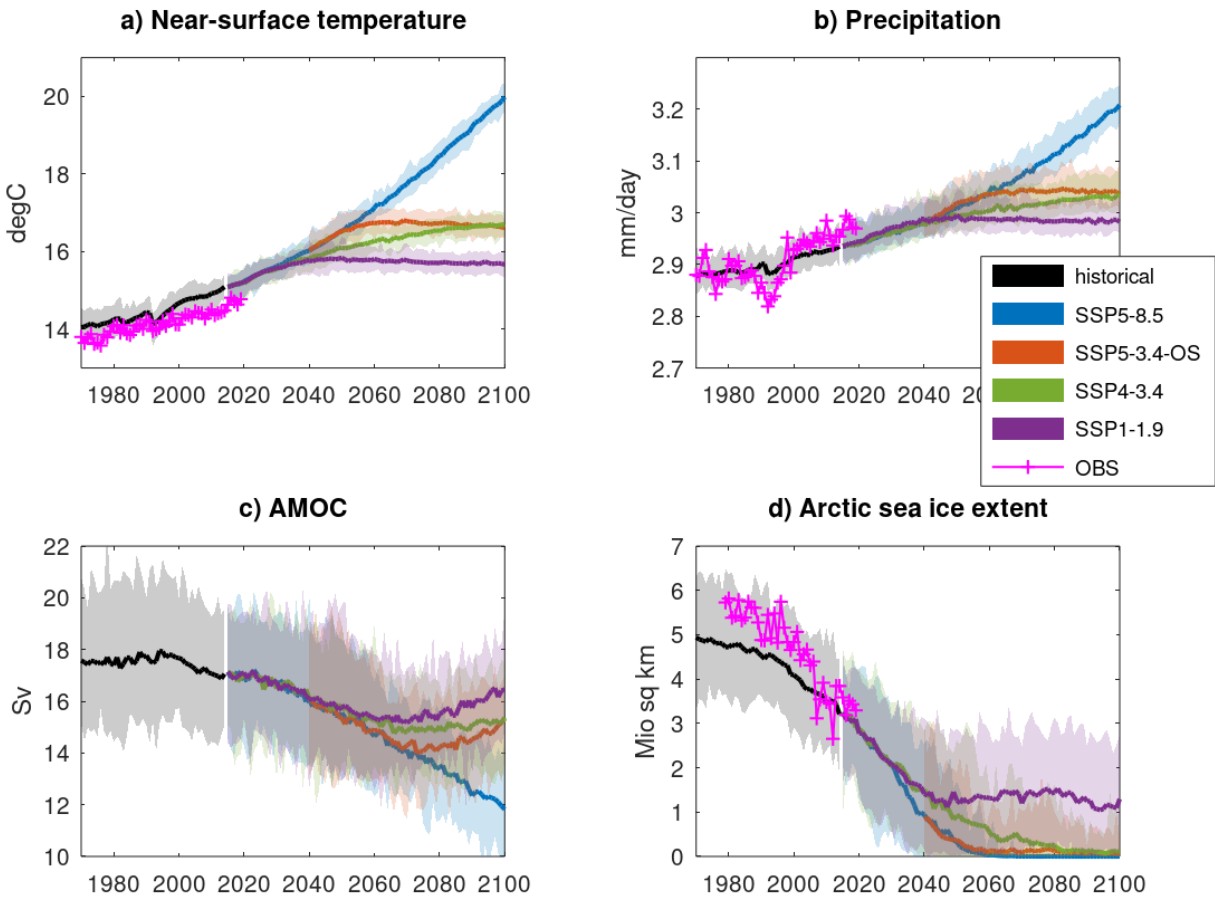

Figure 2: Timeseries of global annual mean near surface temperature, precipitation, AMOC and minimum Arctic sea ice extent in the historical and scenario experiments. Thick lines denote the ensemble means and shaded area the full ensemble width. The scenarios branch off from the historical experiment in 2015 except for SSP5-3.4-OS that branches off from SSP5-8.5 in 2040. The magenta lines marked with plus signs denote the ERA5 re-analysis (Hersbach et al. 2020) for temperature

and precipitation, and the OSI-450 sea-ice observations (Lavergne et al. 2019).

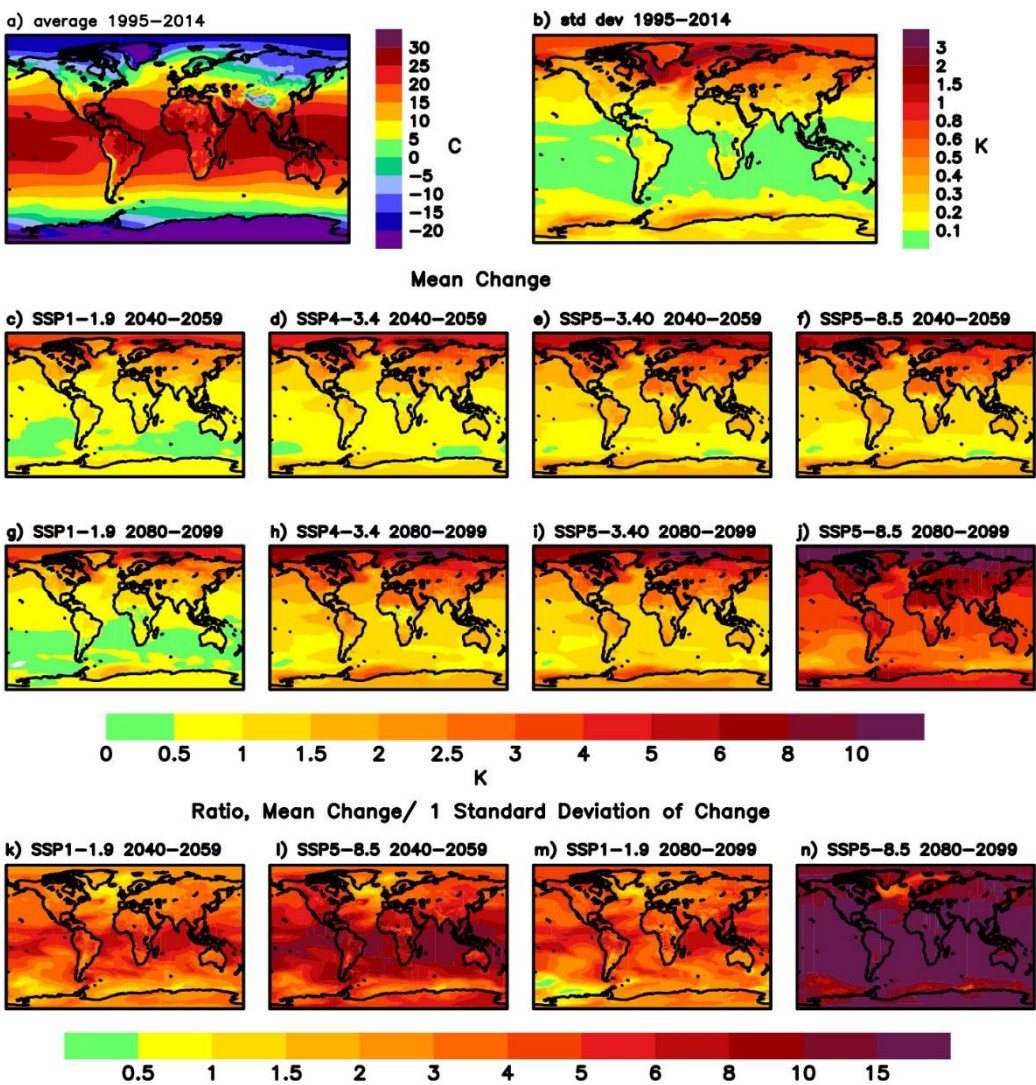

Figure 3: a) Ensemble mean annual mean 2m air temperature, averaged over 1995-2014.

b) One standard deviation of annual mean 2 m air temperature, averaged over 1995-2014, across ensemble members.

c) - f) Ensemble mean 2m air temperature change between 2040-2059 and 1995-2014 for SSP1-1.9, SSP4-3.4, SSP5-3.4-OS and SSP5-8.5. All coloured areas show significant changes at the 95% significance level based on a two-sided student t-test.

g)- j) Same as c) - f) but for changes until 2080-2099.

k) - n) Ratio between mean 2m air temperature change between 2040-2059 (2080-2099 in m and n) and 1995-2014 and one standard deviation of the variability of temperature change across ensemble members in SSP1-1.9 (k and m) and SSP5-8.5 (l, n). The change is calculated for each individual ensemble member as the difference between temperature in the future period (average over 2040-2059 or 2080-2099) and temperature of the reference period (average over 1995-2014) in the same ensemble member.

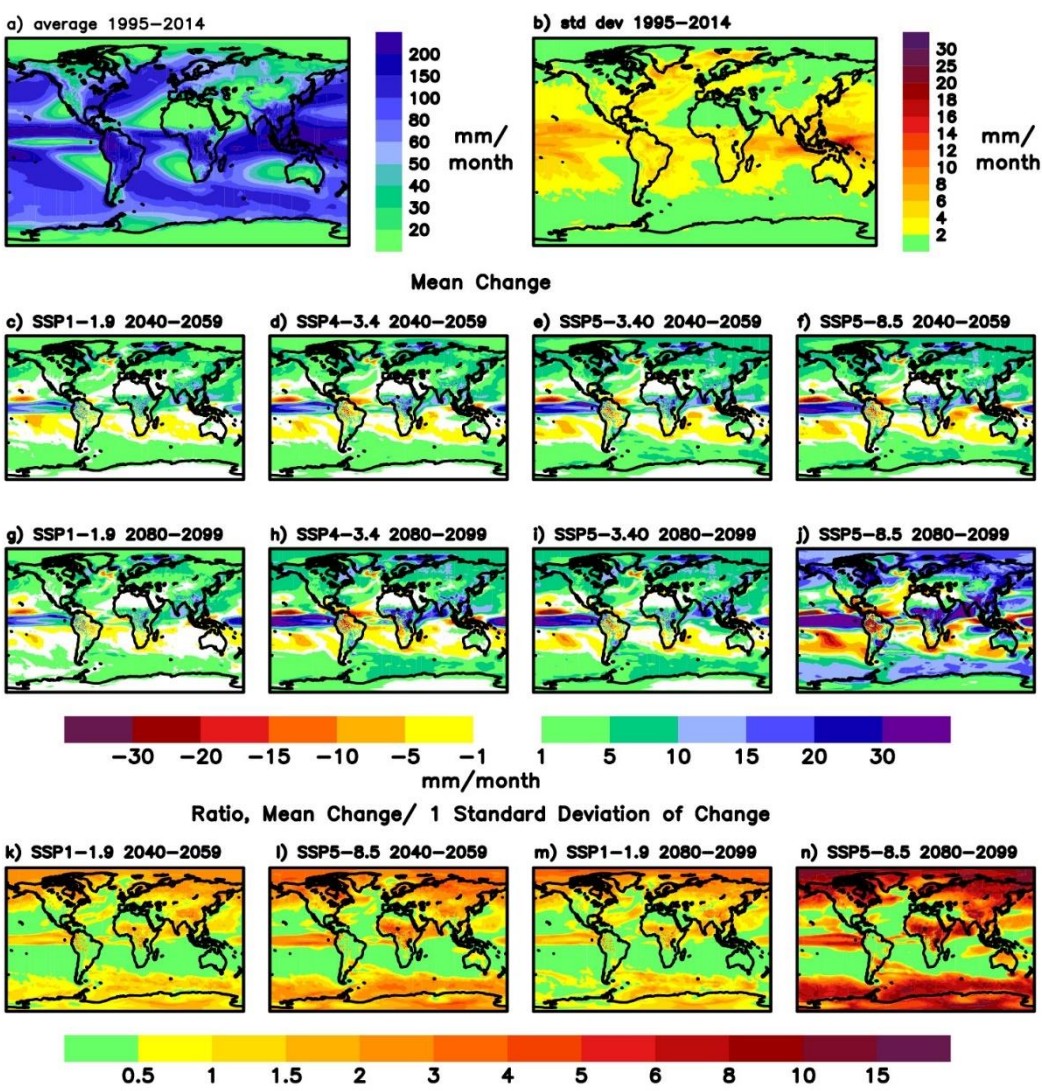

Figure 4: Same as Fig. 3 but for annual mean precipitation. Precipitation changes that are not marked coloured are either not significant at the 95% significance level or small (below +/- 1mm/month).

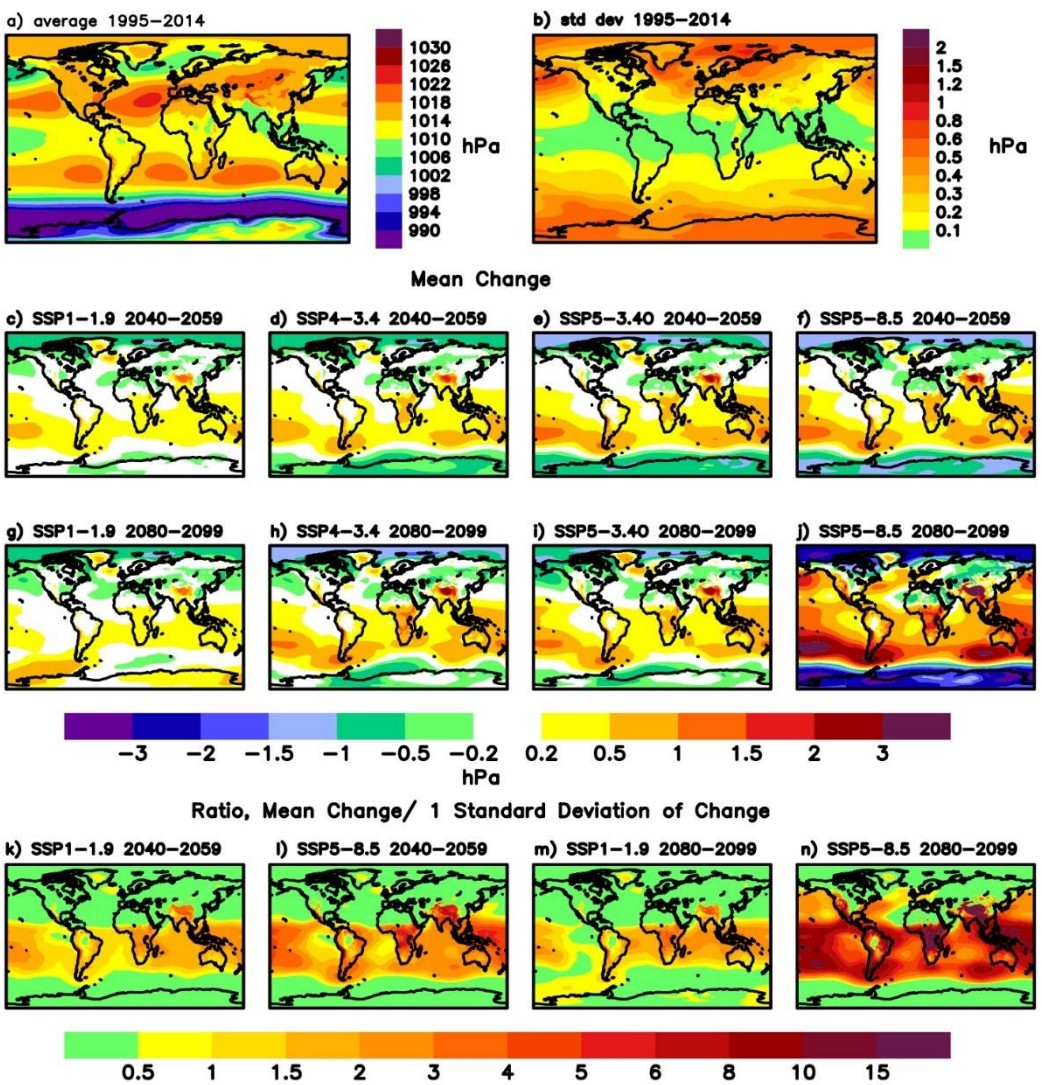

Figure 5: As Fig. 3 but for sea level pressure. Sea level pressure changes that are not marked coloured are either not significant at the 95% significance level or small (below +/- 0.2 hPa).

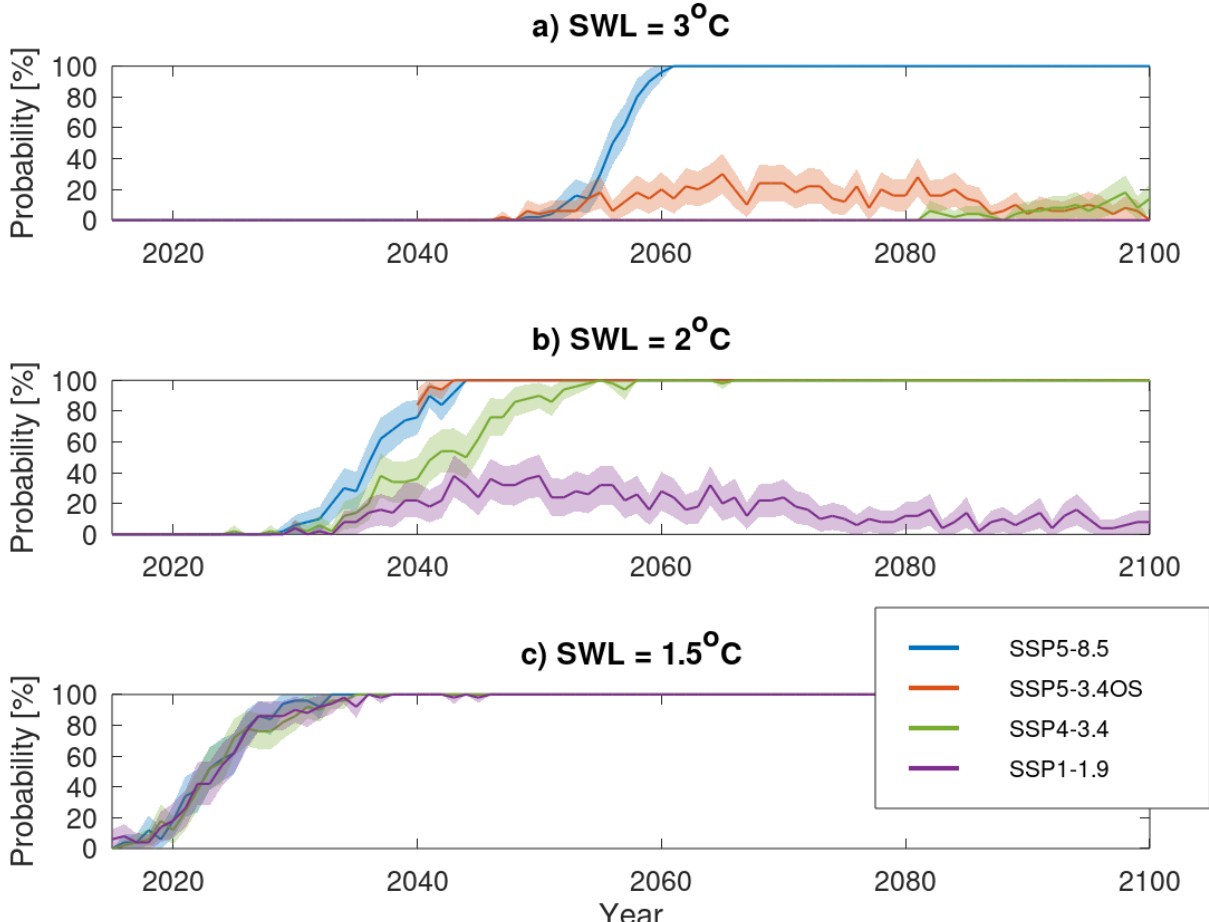

Figure 6: Probability for exceeding a given Surface Warming Level (SWL) in the different scenarios. The SSP5-3.4OS
experiment branches off from SSP5-8.5 in 2040 according to the CMIP6 experimental protocol and thus there are no data for
this scenario before 2040. The shaded area denotes the 95% confidence interval obtained from bootstrapping with 1000
repetitions.

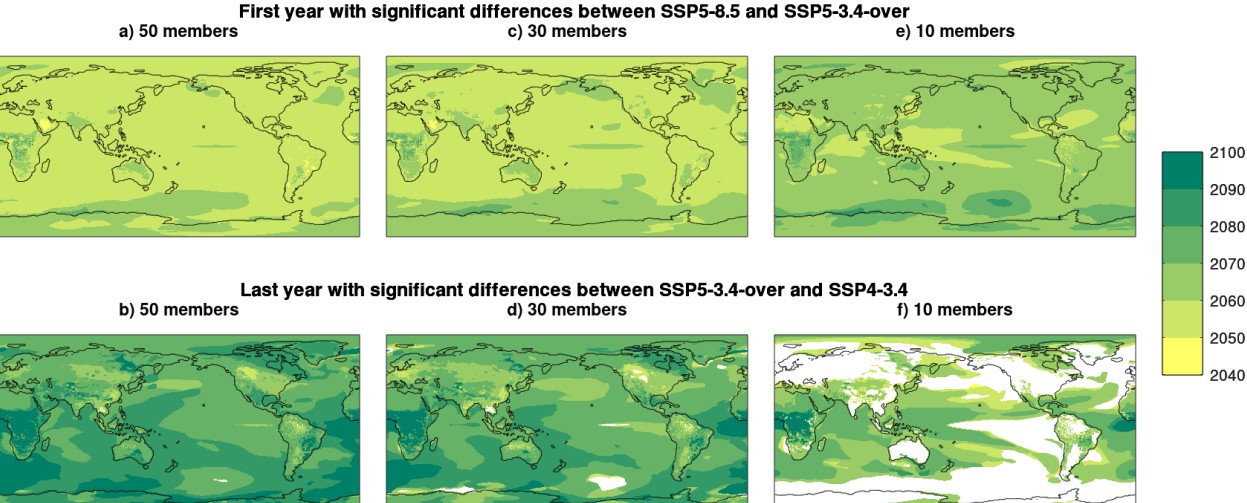

Figure 7: Year of emergence of significant temperature differences between SSP5-8.5 and SSP5-3.4-OS (a,c,e), and the year when the temperature differences between SSP4-3.4 and SSP5-8.5-OS ceases to be significantly different (b,d,f). The results for the full ensemble (50 members) are shown in (a) and (b). The results assuming ensembles with the same ensemble mean and variance are shown in (c) and (d) assuming 30 members, and in (e) and (f) assuming 10 members. White colour denotes regions where the differences between SSP4-3.4 and SSP5-8.5-OS are never significant.