# Peer review of "The SMHI Large Ensemble (SMHI-LENS) with EC-Earth3.3.1"

_Geoscientific Model Development, 2020_

## Author Comment (AC1)

**Reply reviewer 1**

*We'd like to thank Reviewer 1 for his/her careful review of the manuscript and the valuable suggestions for how to improve it. Please find our detailed replies below (in italics).*

This paper describes a new large ensemble with the climate model EC-Earth3 and a relatively unique combination of emissions scenarios. The analysis examples are largely uncontroversial/unsurprising, which is perhaps intentional and adequate for a description paper. The paper is written very clearly, with sound methods and results that support the conclusions. I only have a few small comments that could be considered in a revised version, after which I can recommend publication. I want to express my thanks to the authors for making this data available to the community; it is very valuable.

Three requests as a curious reader:

Could the authors make a few intuitive comparisons with CMIP6? In particular, I was under the impression that EC-Earth3 has among the largest decadal global temperature variability of any CMIP6 model (at least in the piControl). How does this affect its signal/noise ratio in comparison to other CMIP6 models and other available large ensembles?

*Indeed, the large variability of EC-Earth3 compared to other models and its impact on the signal to noise ratio is definitely worth mentioning. We have added a paragraph to the conclusions in which we emphasise that the results presented in this study are valid for EC-Earth3 and could look different for other models.*

It would be interesting to divide the change patterns from the different emissions scenario by global mean temperature, so the pattern become comparable. I actually think that would be a more interesting analysis (and easy to do) than the absolute change maps that are shown here. The authors kind of have to show the absolute change patterns for the description paper, I get that, but we already knew how they would look. I think a comparison of the normalized patterns would be very interesting for the overshoot vs non-overshoot path.

*Thank you for this interesting suggestion. We now have looked at the normalised change for the temperature in the different scenarios and found that the patterns are very similar. Therefore we don't think that it is worth including this result in the paper (attach figure).*

[Figure]

normalized change

a) SSP1−1.9 2040−2059  b) SSP4−3.4 2040−2059  c) SSP5−3.40 2040−2059  d) SSP5−8.5 2040−2059

e) SSP1−1.9 2080−2099  f) SSP4−3.4 2080−2099  g) SSP5−3.40S 2080−2099  h) SSP5−8.5 2080−2099

0.2  0.4  0.6  0.8  1  1.2  1.5  2  2.5  3  4  5

I encourage a comparison with or at least discussion of Sanderson et al. (2017) and several references therein, which investigate what might have been the first "large" (n=10) ensemble of overshoot simulations, albeit with CMIP5 forcing and not CMIP6 forcing.

*We have added a paragraph in the introduction about the Sanderson et al. paper to acknowledge that we are not the first group to look at the effect from an overshoot. However, we cannot do a more detailed comparison as the forcing used by Sanderson et al. is very different from our forcing. Sanderson et al look at overshoot and steadily increasing projections that end with 1.5 deg warming in 2100 which would correspond to a forcing below 2 W/m2 while we look at a substantially larger forcing of 3.4 W/m2 in 2100 and a substantially stronger overshoot around mid-century.*

Other comments:

There have been some concerns with the transition of biomass burning (BB) forcing from the satellite period to the future emissions scenarios in the CMIP6 forcing files. The forcing file has high variability during the satellite era and abruptly lower variability thereafter, at least in certain regions of prominent BB. A large ensemble is perfect to investigate whether this has an appreciable effect on the simulated climate during that transition period. It's not a priori clear whether it affects each model, as aerosol forcing is implemented quite

differently across models. It would be valuable information for the community whether there's any sensitivity to this issue in EC-Earth.

*This is also a very interesting question but we deem it to be outside the scope of the overview paper. Furthermore, it is not clear if this question can be addressed at all with the EC-Earth3 model that doesn't use an interactive aerosol module but instead the simple aerosol plume model MACv2-SP to prescribe the optical properties of aerosols. MACv2-SP doesn't distinguish explicitly between biomass and other aerosol species, it only tells whether a plume is dominated by biomass without telling the exact contribution from biomass.*

Fig. 2: Please add some observations (at least for tas, pr, and sic) as a minimal model validation and to illustrate how the ensemble range compares to real-world variability.

*Thanks for the suggestion, we have added observations/reanalyses for tas, pr and sea-ice to Figure 2.*

Some map colorbars are a bit unintuitive, as they have a color switch offset from zero (e.g., Fig 3c-j). Also, rainbow or rainbow-like colorbars aren't encouraged.

*We have reworked the colorbars in Figs. 3-5 to make them more intuitive. Re rainbow-like color palette: the co-author who has prepared the figures (TK) is heavily colorblind himself and uses these specially designed color tables that allow him to distinguish the different colors while still providing an appealing appearance for non-colorblinds. We therefore don't think that it is necessary to replace the chosen color tables in Figs. 3-5. In Figs 2 and 6 we have replaced the orange color by green to increase the contrast to red. And in Fig.7 we now use a yellow-green color table with increasing intensity. Furthermore, we have also improved the quality of the figures by saving them at higher resolution.*

L188: "mainly due to reduced variability of the change across members" sounds confusing to me. Do the authors just mean "due to the larger signal under stronger forcing"?

*Changed.*

L218: "highly relevant" rather than "highly important"

*Changed.*

L151: that's actually quite interesting. Could the authors speculate what could cause this?

*The oceanic convection in the Labrador Sea is an important source for variability of the AMOC and this convection shows large natural variability on long time scales in EC-Earth. This results in a large spread of the AMOC across model members in the historical period and the early 21st century. Moving towards a warmer climate with increased ocean surface temperatures and a fresher ocean surface in the Labrador Sea, the convection collapses completely in all model members until the middle of the 21st century (Brodeau and Koenigk 2016, Koenigk et al., accepted). Consequently, the spread of convection in the Labrador Sea and thus the spread of the AMOC is reduced. The figure below shows one standard deviation of convection in the Labrador Sea in March across all model members. Here, the ssp5-8.5 scenario is shown but a similar behaviour occurs in the other scenarios as well. We see that most of the reduction happens already before the differences between the different emission scenarios become dominant.*

*We added some discussion to the manuscript to explain the decrease in ensemble spread.*

[Figure]

*Figure: Time evolution of one standard deviation of March mixed layer depth in the Labrador Sea (in m, averaged over 45-72N, 270-310W), calculated for each March as standard deviation across the 50 model members.*

L100: "There are differences"

*Changed.*

L116: "tell us"

*Changed.*

References:

Sanderson, B. M., et al., 2017: Community climate simulations to assess avoided impacts in 1.5 and 2°C futures. Earth Syst. Dyn., 8, 827–847.

---

## Author Comment (AC2)

**Reply reviewer 2**

*We'd like to thank Reviewer 2 for his/her careful review of the manuscript and the valuable suggestions for how to improve it. Please find our detailed replies below (in italics).*

This paper is written as an overview/introduction to the SMHI-LENS. The paper is well written and provides a sufficient introduction to this model. However, the paper misses some relevant literature in the introduction, need some more detail on the initialization of the ensemble and could use minor changes to the Figures to help with interpretation by the reader. I recommend that the paper is revised before it is accepted.

Comments are as follows

Section 1:

While this provides a good introduction, it is unclear why the authors cite specific large ensembles and not others (see line by line comments).

*We mention all models from the list of large ensembles with CMIP6 models on [https://www.cesm.ucar.edu/projects/community-projects/MMLEA/](https://www.cesm.ucar.edu/projects/community-projects/MMLEA/) that have more than 30 members during the historical period, this is clearly stated in the text. We also provide the reference to the MMLEA website which should allow interested readers to find more information.*

The introduction would benefit from a paragraph describing some of the interesting work already done using large ensembles. While the literature is too large to include everything, some references perhaps relating to what is shown later in the manuscript, or a brief introduction to new science done with large ensembles should be included.

*Indeed, there is a lot of work already done with single model large ensembles. A very good source of information is the recent special issue about large ensembles in Earth System Dynamics ([https://esd.copernicus.org/articles/special_issue1037.html](https://esd.copernicus.org/articles/special_issue1037.html)). We have added some examples of previous studies to the introduction.*

Section 2.2 Initial conditions:

Please include the specific years that you used for the initial states in a table.

*We have rephrased the description of how the initial conditions were generated and added Table 1 that links the historical CMIP6 experiments with the initial conditions for the large ensemble.*

Figures: are rainbow colorbars the best choice? Perhaps you can find a better colorbar

*The co-author who has prepared the figures (TK) is heavily colorblind himself and uses these specially designed color tables that allow him to distinguish the different colors while still providing an appealing appearance for non-colorblinds. We therefore don't think that it is necessary to replace the chosen color tables in Figs. 3-5. In Figs 2 and 6 we have replaced the orange color by green to increase the contrast to red. And in Fig.7 we now use a yellow-green color table with increasing intensity. Furthermore, we have also improved the quality of the figures with higher resolution to improve legibility.*

F2 – poor quality and fuzzy

two orange colors are difficult to distinguish by eye on my computer screen

*Indeed, we apologize for the poor quality of F2 in the first version. We have now improved the quality and replaced the orange color with green.*

F6 – b) the orange line seems to come from nowhere

*The red (previously orange) line is for ssp534-os that branches off from ssp585 in 2040 following the CMIP6 protocol, therefore there are no data for ssp534-os before 2040.*

c) I don't see the orange line at all

perhaps different symbols or dots, dashes could be used so we can see all colors

*The scenarios are very similar until about 2040 (see Fig. 2) and all of them pass SWL 1.5 during this period. Therefore they all lie on top of each other and it is difficult to distinguish between them. We have added an explanation to the text to make this clear.*

F7 – Please describe in the caption how you compute the 10 member result. Do you pick one set of 10 members or resample 10 members many times?

*We explain in the text that we use a hypothetical ensemble with the same mean and variance as the large ensemble but only 10 members. With this approach no 10-member samples have to be picked, the question we try to answer is simply what would change if we had a similar ensemble in terms of mean and variability but only 10 members.*

Given most large ensembles have 30 members, as you note in your introduction. It would be good to do this for 30 members as well as 10 members and add a panel to the Figures.

*Thank you for the suggestion, we have added similar plots for a hypothetical 30 member ensemble. The results are interesting and show that there isn't much impact from reducing the sample size from 50 to 30, but quite some impact when reducing the sample size even further to only 10 members.*

Would it be worth considering precipitation for these Figures as well given the pathway dependence of this variable:

e.g https://journals.ametsoc.org/view/journals/clim/30/11/jcli-d-16-0441.1.xml

https://agupubs.onlinelibrary.wiley.com/doi/full/10.1002/2016GL070869

https://agupubs.onlinelibrary.wiley.com/doi/full/10.1029/2018JD028821#:~:text=We%20find%20a%20robustly%20larger,GHGs%20across%20all%20available%20models.&text=This%20is%20because%20of%20a,by%20the%20GHG%20atmospheric%20forcing.

*We have looked at precipitation and found that the differences between the scenario ensembles are not significant, and therefore do not include this figure here.*

Line by line comments:

Line 34- This is also shown using large ensembles in the following two papers:

https://esd.copernicus.org/articles/11/491/2020/

https://iopscience.iop.org/article/10.1088/1748-9326/ab7d02/pdf – this could also be compared to the results on line 185-186 in the discussion

*Thank you for pointing us to these interesting references that show how internal variability is obscuring the climate change signal in the near and partly even mid-term future. We have added these references to the introduction. However, we don't think Maher et al (2020) fits well with the discussion in Section 3.2 because they study the warming in the near-future while our plots show the warming in the middle and at the end of the century.*

Line 46 – MPI-GE is not MPI-ESM-LR but MPI-ESM1.1 – additionally the correct acronym for this large ensemble is MPI-GE not MPI-ESM-GE

*Corrected.*

Line 46 – I am confused about the choice of models introduced here. The large ensemble archive introduced by Deser et al 2020 includes more models, why not introduce all of the ones in this archive?

*See reply above.*

Line 51 – Also GFDL-SPEAR is now available online:
https://agupubs.onlinelibrary.wiley.com/doi/10.1029/2019MS001895

*Thanks for mentioning this additional ensemble that is not mentioned (yet?) on the MMLEA website, we have added it to the list of large ensembles in the introduction.*

Line 70 – RCM large ensembles already exist. It would be worth citing these here:

https://journals.ametsoc.org/view/journals/apme/58/4/jamc-d-18-0021.1.xml

*We have added a sentence with references to already existing studies with large ensembles of regional climate models.*

106 – is there a citation for SSPs and ScenarioMPI?

*Yes, we cite the ScenarioMIP paper (O'Neill et al 2016) in the introduction where we describe our choice of scenarios.*

155- I believe this is usually called TAS? Would it be more understandable to use the standard acronym – also please be consistent as you use tas in Figure2's caption

*We have changed T2m to tas, P to pr and SLP to psl for consistency throughout the manuscript.*

163 – perhaps 3K and higher is better wording

*Changed.*

163 – 'the' northern hemispheric

*Added "the".*

173 – is increasing → 'increases'

*Changed.*

177 – should this be 'divided by'?

*Changed.*

221 – it would be interesting to add whether the Aleutian low is too pronounced in all ensemble members as we would not expect observations to agree with the ensemble mean. This applies for all the metrics discussed on these lines.

*The std deviation across the ensemble is considerably smaller than the biases in the Aleutian low and other areas (Fig 5b). Therefore we think the bias is a robust feature of the ensemble in the sense that most ensemble members have a bias of the same sign. We added a sentence to the text.*

236 – however this result contrasts with the following work, which should be added on this line

https://journals.ametsoc.org/view/journals/clim/30/11/jcli-d-16-0441.1.xml

*Thank you for pointing us to this work, we have modified the sentence accordingly and added the reference.*

509 – specify what the nino3.4 region is

*The definition of the Nino3.4 region from https://climatedataguide.ucar.edu/climate-data/nino-sst-indices-nino-12-3-34-4-oni-and-tni has been added.*

---

## Author Response (AR2)

Hej Sophie---

Thank you for your valuable comments and suggestions. Please find our replies below in italics. We have also updated the manuscript accordingly.

Kind regards,
Klaus

My main remark concerns the need to clarify the number of members. In Table 1, you show 9 starting dates for each of the 6 historical simulations: this would make 54 members and not 50; can you clarify?

*We have added* "The branch_time 1974-1-1 is only used for r1 and r2 of the CMIP6 historical experiment, yielding members 149 and 150 of the large ensemble." *to the caption of table 1 which hopefully explains how we get the 50 members.*

And also, I suppose that each of the 50 members run for the period 1970-2014 is continued for each of the 4 scenarios mentioned, i.e. you have 50 members for each scenario, right? Can you clarify this up front in section 2.3, I had to read until section 3.4 to be fully sure that this is the case.

*We have added* "Each of the scenarios comprises 50 members that start from the end of the corresponding member of the historical experiment." *to the 1st paragraph of Sec. 2.3.*

Regarding reviewer 1's suggestion to provide normalised patterns, you reply that it is not worth including them in your paper as they are very similar to the non-normalised ones. To show this, you provide the figure of the normalised patterns. I would say that your conclusion is not convincing, as: 1- the color scales of the two figures are not the same (one goes from 0.2 to 5 while the other goes from 0 to 10); 2- the two plots for SSP5-8.5 2080-2099 look quite different to me. Can you clarify and further argue why you think it is not appropriate to include the normalised patterns?

*In the reply to Reviewer 1 we have included a figure showing the normalized temperature changes for 4 scenarios (rows) and 2 different time periods in the future (columns). We find that the patterns of normalised changes are very similar across all scenarios (for each time period), and therefore we decided to not include this figure in the manuscript. By no means is this figure comparable to Fig. 3 of the paper that was never intended. Our point is that normalised patterns are similar across scenarios and therefore do not contribute much to the discussion.*

And I have the following technical points:

- In the title, you have to mention the version of EC-Earth3 used so please add "3.3.1"

*Done.*

- Please define MPI-GE the first time it appears on p.2. Currently, it is defined on p.12.

*Done.*

- Please use "Fig." everywhere to refer to figures.

*Done.*